# Environmental impacts, water footprint and cumulative energy demand of match industry in Pakistan

Najeeb Ullah[1☯], Syeda Asma Bano[2☯], Ume Habiba[1☯], Maimoona Sabir[2☯], Andleeb Akhtar[3☯], Samreen Ramzan[4☯], Ayesha Shoukat[4☯], Muhammad Israr[5☯], Sher Shah[1☯], Syed Moazzam Nizami[1☯], Majid Hussain[1☯]*

1 Department of Forestry and Wildlife Management, University of Haripur, Haripur, Khyber Pakhtunkhwa, Pakistan, 2 Department of Microbiology, University of Haripur, Haripur, Khyber Pakhtunkhwa, Pakistan, 3 Department of Psychology, University of Haripur, Haripur, Khyber Pakhtunkhwa, Pakistan, 4 Department of Commerce, The Islamia University of Bahawalpur, Punjab, Pakistan, 5 Department of Biology, University of Haripur, Haripur, Khyber Pakhtunkhwa, Pakistan

☯ These authors contributed equally to this work.
* majid@uoh.edu.pk

**Data Availability Statement:** All relevant data are within the manuscript and its Supporting Information files.

## Abstract

A comprehensive life cycle assessment (LCA) was conducted for the matchsticks industry in the Khyber Pakhtunkhwa province of Pakistan to quantify environmental footprint, water footprint, cumulative energy use, and to identify improvement opportunities in the matchsticks manufacturing process. One carton of matchsticks was used as reference unit for this study. Foreground data was collected from the matchsticks industry through questionnaire surveys, personal meetings, and field measurements. The collected data was transformed into potential environmental impacts through the Centre for Environment Studies (CML) 2000 v.2.05 method present by default in the SimaPro v.9.1 software. Water footprint was calculated using methodology developed by Hoekstra et al., 2012 (water scarcity index) V1.02 and cumulative energy demand by SimaPro v.9.1 software. The results showed that transport of primary material (wood logs), sawn wood for matchsticks, red phosphorous, acrylic varnish, and kerosene fuel oil contributed to the overall environmental impacts. Transport of primary materials and sawn timber for matchsticks contributed significantly to abiotic depletion, global warming, eutrophication potential, ozone depletion, corrosion, human toxicity, and aquatic ecotoxicity effects. The total water footprint for manufacturing one carton of matchsticks was 0.265332 $m^3$, whereas the total cumulative energy demand was 715.860 Mega Joules (MJ), mainly sourced from non-renewable fossil fuels (708.979 MJ). Scenario analysis was also conducted for 20% and 30% reduction in the primary material distance covered by trucks and revealed that reducing direct material transport distances could diminish environmental impacts and energy consumption. Therefore, environmental footprint could be minimized through diverting matchsticks industries freight from indigenous routes to high mobility highways and by promoting industrial forestry close to industrial zones in Pakistan. Many industries did not have emissions control systems, exceeding the permissible limit for emissions established by the National Environmental

**Funding:** The author(s) received no specific funding for this work.

**Competing interests:** The authors have declared that no competing interests exist.

**Abbreviations:** AD, Abiotic depletion; AP, Acidification potential; CED, Cumulative energy demand; CO2, Carbon dioxide; CORRIM, Consortium for Research on Renewable Industrial Materials; CVw, Weighted Coefficient of Variation; EP, Eutrophication potential; EPA, Environmental protection authority; FAE, Freshwater aquatic ecotoxicity; GHS, Greenhouse gas; GWP, Global warming potential; Kg, Kilogram; KWh, Kilo Watt Hour; LCA, Life cycle assessment; LCI, Life cycle inventory; LCIA, Life cycle impact assessment; MAE, Marine-water aquatic ecotoxicity; MJ, Mega joule; OLD, Ozone layer depletion; PO, Photochemical oxidation; TE, Terrestrial ecotoxicity; WSI, Water scarcity index.

Quality Standards (NEQS) of Pakistan. Thus, installation of emissions control system could also diminish emissions from match industry in Pakistan.

## Introduction

Matchsticks are generally produced from small wooden sticks or stiff paper [1]. One end of the matchsticks is coated with crude material ignited by frictional heat brought about by rubbing the matchsticks' coated end against the matchsticks box's contrary side [2]. The matchsticks' coated end is known as the matchsticks head and comprises a binder and chemicals. Safety matchsticks can rub easily with any appropriately frictional surface [3]. Raw materials used in the manufacture of safety matchsticks include softwoods, paper boxes, and chemicals for the matchstick's heads and rubbing the packages' side includes splints, veneers, wax, chlorate, sulfur, bichromate of potassium, resin, gums, glass powder, glue, phosphorous, kerosene oil, linseed oil, ammonium phosphate, potassium chlorate [4]. Safety matchsticks are one of the oldest wood-based industries globally. The industry has played an essential role in the world's domestic and national economy by providing value supplements to the wood-based industry, employment generation, and local community's livelihood in Pakistan. Matchsticks have been considered as one of the most important daily use commodities by common-man. The majority of low-income families living in rural areas depend on matchsticks to ignite their lighting lamps, cooking stoves, and burners. Moreover, smokers used safety matchsticks to ignite cigarettes, hookas, and beedies in Pakistan [5], whereas matchsticks are used in different religious ceremonies and worship places such as mosques, churches and temples [6].

Pakistan is one of the leading manufacturers of safety matchsticks globally. The country has been trading matchsticks since 1990s to a variety of world markets comprising Africa, the Middle East, Europe, and Latin America. Revenue produced from safety matchsticks exports surpasses 40 million US dollars each year in Pakistan [7]. The matchsticks industry in Khyber Pakhtunkhwa (KP) province of Pakistan avails the subsidies of locally delivered raw materials and inexpensive labor costs. The large wood production deriving from the region's poplar (*Populus*) trees is fulfilling the matchsticks industry's needs. Most basic production units of matchsticks sector are located in KP province of Pakistan, ranked third position in the global export of safety matchsticks in 2014 with a worth of US$ 18.5 million while India and Sweden ranked first and second in the worldwide export of security matchsticks in 2014 with a value of US$ 46.7 and 36.5 million, respectively [8]. There are more than twenty-five (25) matchsticks industries operating in KP, Pakistan. These industries are manufacturing thousands of matchsticks box cartons every year. Consequently, enormous quantities of chemicals and energy were used in the manufacturing process of matchsticks products in Pakistan. The consumption of primary and secondary materials, such as wood logs, chemicals, energy use, and transportation activities, leads to the production and the emission of toxic and greenhouse gases (GHG), which causes severe environmental impacts [9]. Several activities, such as chemical mixing, chemical grinding, box filling, matchsticks packing [10], including hazardous chemicals use such as potassium chloride, potassium bichromate, zinc oxide, sulfur, red phosphorous, different types of glues, kerosene, and linseed oil, inks, and colours. These chemicals result is detrimental impacts and causes environmental hazards related to the matchsticks industry in Pakistan [11]. Therefore, these environmental hazards need to be investigated.

Life Cycle Assessment (LCA) is a tool that quantify environmental impacts by measuring the energy and materials used, and wastes thrown into the environment [12–14]. LCA is a

holistic, structured, universally accepted tool for assessing emissions and associated environmental and human impacts posed by processes or products throughout their life cycle stages [15]. The steps of the life cycle deemed may comprise resource production ("cradle") through material manufacture and assembly ("gate"), use, recovery, reuse, and demolition ("grave") [13, 16]. In developed countries, industries conduct LCA-based research for their products to comply with the National and International permissible limits of contaminants emission to water, air, and soil and lower greenhouse gas (GHG) emissions produced [17]. However, there was no precise and reliable data of matchsticks industries in KP, Pakistan, and no published LCA regarding matchsticks products. Therefore, the present study's objectives were first, to collect information on matchsticks manufacture and determine the material supply chain, energy consumption, outflows to the soil, water, and air, during the manufacturing process in KP, Pakistan. Second, to estimate different potential environmental impacts, such as global warming potential (GWP), abiotic resource depletion (AD), ozone depletion, eutrophication potential (EP), photochemical oxidation (PO), acidification potential (AP), terrestrial Ecotoxicity (TE), aquatic Ecotoxicity (MAE), freshwater aquatic Ecotoxicity (FAE), and human toxicity (HT), water footprint and cumulative energy use and third, to improve the matchsticks industry eco-efficiency by identifying alternative production scenarios.

## Materials and methods

### Study area

This study was conducted in three (03) bigger industrial zones i.e., Hattar Industrial Estate—District Haripur, Hayatabad Industrial Estate—District Peshawar, and Special Industrial Zone Risalpur—District Nowshera of Khyber Pakhtunkhwa province of Pakistan as shown in (Fig 1) District Haripur is situated in the Hazara region of Khyber Pakhtunkhwa, located in the country's northeastern region [18]. Peshawar is the provincial capital of Khyber Pakhtunkhwa and has been known for quite some time as 'Frontier Town' standing right at the passage of the world-famous Khyber Pass [19]. Nowshera is the name of the region, which is a mix of two Persian words, "Now" and "Sheer," altered into Nowshera with time, meaning "new city" [20].

### Study design: Life cycle model and inventory

**Goal and scope definition.** In the present study, energy, material, and emission flows have been calculated based on one carton matchsticks produced from June 2019 to June 2020 in KP, Pakistan. Therefore, one carton of matchsticks was considered as functional unit to accomplish this study's goal. The cradle-to-gate life cycle inventory (LCI) approach was applied for one carton matchsticks production in KP, Pakistan.

**The system boundary of the study.** The system boundary of the present study is presented in (Fig 2). The detailed steps for matchsticks production are given below:

A. *Logs transportation to matchsticks industries*. The wood logs harvested from the agroforestry plantations are transported by medium-sized trucks to matchsticks industries and stored in the wood log yard where logs were debarked. Data concerning the fossil fuels source consumed for materials transport to matchsticks industries were collected from the matchsticks managers and transporters.

B. *Debarking*. The bark is considered waste or residue in matchsticks manufacture, as it is generally removed before matchsticks production. Some industries use the bark material, branches, and knots in other internal processes while others sold it to the local community as firewood.

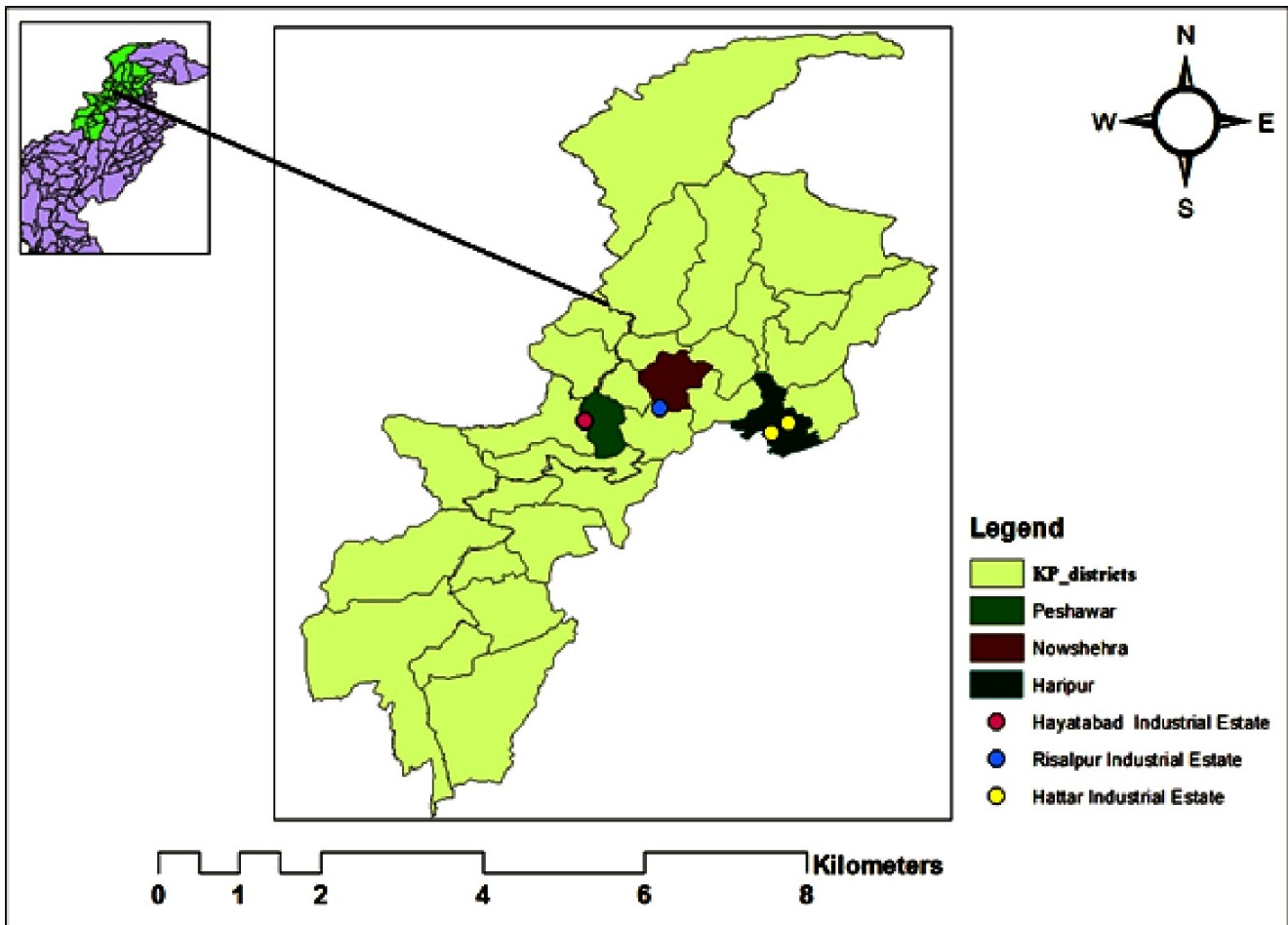

**Fig 1. Location map of the study area.**

C. *Splints and veneers*. After debarking, the slings are produced for making sticks for match, which are used for holding purposes. It is made of softwood. Usually, the Poplar tree species have splints in the matchsticks manufacturing industry in KP. Simultaneously, the veneers used for making matchstick boxes are also made out of softwood trees.

D. *Chemical dipping*. This step starts with the preparation of a combination of various chemicals with distinct ratios. The chemical mixture is made of two types. The first type is for the head of the matchsticks, while the other is for the sides of matchstick boxes used for a striking matchstick. The producers mix various classes of chemical based on their needs. The glue solution's preparation is precious because it is used as a binding agent in matchsticks manufacture. After dipping, the matchsticks sticks were placed for cooling.

E. *Box making*. Box-making is the process of matchsticks box production or packing. The box is the pouch of matchsticks sticks and is also utilized for stick striking. The box-making process can be separated into two types, i.e., outer compartments and inner boxes.

F. *Box filling*. The dried matchsticks are filled in the inner box and incorporated along with the inner container into the outer carton. The number of sticks to be filled in a matchstick box depends on matchstick box size such as small size (45×30 ×14 mm) contain 30–35

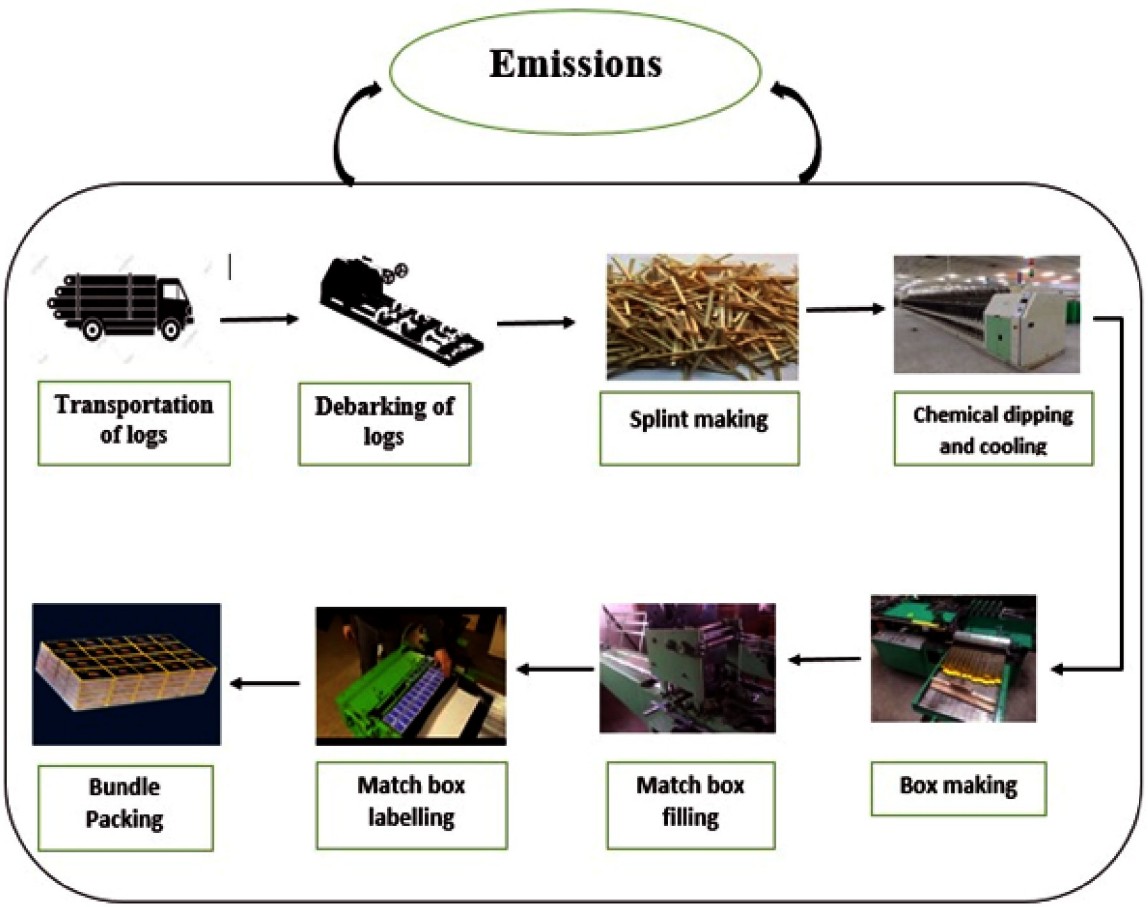

**Fig 2. Cradle-to-gate system boundary of the study.**

splints, and medium-size (51×36×14.5 mm) include 40–45 splints while large size (60×45×15 mm) has 60–65 slings.

G.  *Band rolling and label.* Band rolling and labels are the two activities made instantaneously. This step gives the final touch to matchsticks production. The labeling is done directly after band rolling, which follows box filling and side chemical varnish. The matchsticks industry must wrap a band roll strip over each matchstick box, around its open end. Labeling is a method by which the trade labels are pasted over the band roll ends, which is needed to tear off the band roll to open the matchstick box. The trade labels are printed pieces of paper, including the name and symbol of the product, producer, place of manufacture, and the like to identify the matchsticks' producers.

H.  *Bundle packing.* It is the last step involved in matchsticks manufacture. Matchstick boxes must be well packed. There are three types of packing, namely one dozen packing (10 matchstick boxes), gross packing (100 matchstick boxes), and bundle packing (1000 or 500 matchstick boxes). After the packaging, the products are sent to storehouses through medium or small vehicles.

**Life cycle inventory.**    Eight (08) matchsticks industries were visited and data were collected from the matchsticks industry managers and laborers (Table 1). Five (05) matchsticks

**Table 1. Name and address of the selected match industries visited for LCI for the present study.**

| Name of industries | Address |
|---|---|
| 1) Pine Match (Pvt) Ltd | Factory; plot#51/3, phase2, Hattar Industrial Estate, Khyber Pakhtunkhwa. Phone No: 0995617032 |
| 2) Syed Match Company Limited | Rehana Road, Sarai Saleh Tehsil Haripur, Khyber Pakhtunkhwa. Phone No: 0995319189 |
| 3) Alam Match (Pvt) Ltd | Plot#109, Hayatabad Industrial Estate, Jamrud Road Peshawar, Khyber Pakhtunkhwa. Phone No: 0915823001 |
| 4) Venus Match Industry (Pvt) Ltd | Plot No #97, F-1, Industrial Estate, Jamrud Road Peshawar, Khyber Pakhtunkhwa. Phone No: 0915814901 |
| 5) Khyber Match Factory (Pvt) Ltd | Plot#10, Industrial Estate, Jamrud Road Peshawar, Khyber Pakhtunkhwa. Phone No: 0915812813 |
| 6) Ashraf Match (Pvt) Ltd | Plot#30-34/W, Industrial Estate, Jamrud Road Peshawar, Khyber Pakhtunkhwa. Phone No: 0915813951 |
| 7) Mohsin Match Factory (Pvt) Ltd | Plot#90-B, Industrial Estate, Jamrud Road Peshawar, Khyber Pakhtunkhwa. Phone No: 0915815056 |
| 8) Pakistan Match Industries (Pvt) Ltd | Plot #152, Special Industrial Zone Risalpur, Nowshera, Khyber Pakhtunkhwa. Phone No: 092388026 |

industries were visited in the Hayatabad Industrial Estate, Peshawar, and two (02) in the Hattar Industrial Estate, Haripur, while one (01) matchstick industry was visited in the Special Industrial Zone at Nowshera, KP, Pakistan. Production-weighted average values were calculated for the primary data collected from the eight-matchstick industries. The coefficient of variation (CV) was calculated, which defines the variability of the data by dividing the standard deviation (SD) by the mean [14] using Eq 1. The weighted SD was calculated by Eq 2. Moreover, the $CV_w$ was calculated for each input by using Eq 3 as given below.

$$\bar{x}_w = \frac{\sum wx}{\sum w} \tag{1}$$

$$Sd_w = \sqrt{\sum_{i=1}^{N} w_i (x_i - \bar{x}_w)^2 \, x \, \frac{N'}{(N'-1)\sum_{i=1}^{N} w_i}} \tag{2}$$

$$CV_w = \frac{Sd_w}{\bar{x}_w} \tag{3}$$

Secondary data were collected from the internet, literature, published research articles, and database such as Eco-invent and Consortium for Research on Renewable Industrial Material (CORRIM).

**Life cycle impact assessment.** The data collected for life cycle inventory were assessed in SimaPro software v 9.2, applying the CML 2000 model, for the following impacts: (AD), (GWP), (OLD), (EP), (AP), (PO), (MAE), (TE), (FAE), and (HT). For energy consumption, the cumulative energy demand indicator, present by default in the SimaPro v.9.2 software was applied to estimate the total energy used from various sources. Whereas, the Hoekstra et al., 2012 method was applied to calculate the water footprint caused by one carton of match produced in KP, Pakistan.

## Other assumptions and cutoff rules

Following product category rule "PCR" recommendations, which states that if a product or energy is less than 2% of cumulative energy or mass of the overall, it could be excluded from

the study if its environmental impacts are negligible [21]. The data collection, assumptions and life cycle impact assessment followed the protocol established by the CORRIM for achieving life cycle inventories on wood products [22] and International Organization for Standardization [23]. However, where accurate and reliable data was not available then some assumptions were taken which are below:

- Eight matchsticks' industries were considered state of the art and representative of the Pakistani matchsticks production industries.

- Production-weighted data for each inputs and outputs from eight matchstick industries was considered rather than general overviewed production values of inputs and outputs for the year 2019–2020.

- The logs utilized in the matchsticks manufacturing are assumed to be cut manually and sprang physically using crosscut saws; hence, no energy was consumed in getting wood logs for matchsticks production.

- Primary raw materials were carried by medium trucks with a net load of 160.4 metric tons, while secondary raw materials were taken by large trucks with a load of 2.40 metric tons.

- Diesel fuel was supposed to be utilized for raw materials and product supply. Also, medium-size vehicles consume 1 L of diesel per 10 km during travel and large automobiles consume 1 L of diesel per 6 km during road travel.

## Results and discussion

The life cycle inventory data for one carton of match production by match industries in KP during 2019–2020 are shown in (Table 2). The life cycle impact assessment of one carton of match production and its associated environmental impact categories per process are given in (Fig 3). Transport of primary materials, sawn wood for a match, cobalt, red phosphorous (phosphate rock), dry and resin gum (epoxy resin), acrylic varnish, dry paint (alkyd paint white), and kerosene oil had maximum contributions to most of the impact categories. The on-site industrial processes of the match manufacturing were responsible for most of the impacts in Abiotic depletion, mostly from the transport of primary resource material (81%), sawn wood for a match (10%), kerosene oil (3%), dry and resin gum (epoxy resin) (2%), paraffin wax (2%) and acrylic varnish (2%). In Acidification potential impact category, transport of primary materials has the highest contribution (65%), sawn wood for a match (17%), cobalt (4%) whereas dry and resin gum (epoxy resin) and dry paint (alkyd paint white) contribute (3%). Besides, red phosphorous (phosphate rock) secondary transport also contributes (2%). In Eutrophication potential impact category, again transportation of primary materials had the highest contribution (57%), sawn wood for a match (19%), acrylic varnish and cobalt, each one contributes (5%), red phosphorous (phosphate rock) and dry paint (alkyd paint white) respectively contributes (4%), while paper used (kraft paper) and printing ink mutually contribute (2%).

The transport of primary materials is accountable for maximum contribution (85%) to global warming potential impact category and sawn wood for a match throughout the KP which added about (11%) to the impacts in GWP impact category, while dry and resin gum (epoxy resin) and acrylic varnish equally contribute (2%). The transport of primary materials and sawn wood for a match had a significant contribution (57% and 35%, respectively) to the photochemical oxidation impact category. This significant contribution leads to the emissions of $CH_4$, $CO_2$, CO, $N_2O$ from fossil fuels consumed to transport primary materials. In

**Table 2. Life Cycle Inventory for one carton of matches' production by match industries in KP during 2019–2020.**

| Inputs/output | Unit | Quantity/Amount | Weighted coefficient of variation (CVw)% |
|---|---|---|---|
| Inputs (Materials and energy resources and distances covered) | | | |
| Wood Log | kg | 20 | 57.1% |
| Paper Waste | kg | 2 | 31.60% |
| Electricity | kWh | 0.02 | 39.60% |
| Fossil Fuels (Diesel) | L | 0.14 | 26.70% |
| Paper used | kg | 0.15 | 70.40% |
| Potassium Perchlorate | kg | 0.03 | 95.50% |
| Glass Powder | kg | 0.10 | 79.70% |
| Zinc Oxide | kg | 0.03 | 93% |
| Sulfur | kg | 0.03 | 71.20% |
| China Clay (Kaolin) | kg | 0.04 | 63.40% |
| Boric Acid | kg | 0.003 | 104.90% |
| Colours (Sky blue and Acid violet) used | kg | 0.006 | 100.90% |
| Potassium dichromate | kg | 0.001 | 81.70% |
| Pearl and SWD glue (Adhesive) | kg | 0.07 | 91.50% |
| Dry and Resin gum (Epoxy) | kg | 0.108 | 71.40% |
| Linseed oil (Preservative) | L | 0.005 | 76.70% |
| Kerosene oil used | L | 0.354 | 69.30% |
| Paraffin Wax | kg | 0.22 | 16.40% |
| Red phosphorous (Phosphate rock) | kg | 1.16 | 76.50% |
| Dry Paint (Alkyd paint white without water) | kg | 0.1 | 71.20% |
| Ink | kg | 0.09 | 72.70% |
| Varnish | L | 0.32 | 68.30% |
| Cobalt | kg | 0.05 | 75% |
| Wood Log Distance Travelled | t.km | 160.4 | 47.20% |
| Chemicals Distance Travelled | t.km | 2.4 | 0.80% |
| CNG used In Labour Transportations | kg | 0.04 | 45.2% |
| Co-Products | | | |
| Bark Waste | kg | 2 | 32.90% |
| Wood Waste | kg | 1 | 43.60% |
| Output: One carton of match produced | | | |

comparison, acrylic varnish and dry paint (alkyd paint white) contribute (5% and 3%, respectively). The transport of direct materials and sawn wood were the major contributors (71% and 21%, respectively) to the Human toxicity impacts category. The highest contribution from the transport of primary materials was CO, $CO_2$, and $NO_X$ from the fossil fuels used in the transportation of primary materials in the match production process. Therefore, primary materials transportation was the major contributor to the HT impacts categories. The cobalt and acrylic varnish equally contribute (3%), whereas dry paint (alkyd paint white) contributes (2%) to HT impacts. The primary materials transport account for (87%) of the contributive emissions to ozone layer depletion (OLD), followed by sawn wood for a match (10%) and kerosene (3%), which contributed to the ozone layer depletion (OLD) impact category.

The match manufacture was also responsible for most of the impacts in freshwater aquatic ecotoxicity (FAE), mainly from the transport of primary resource material (66%) and the sawn wood for a match (17%). The acrylic varnish and dry paint (alkyd paint white) contribute (7% and 4%, respectively) while cobalt, potassium perchlorate, and red phosphorous (phosphate rock) contribute together to the freshwater aquatic ecotoxicity impacts category (2%). The

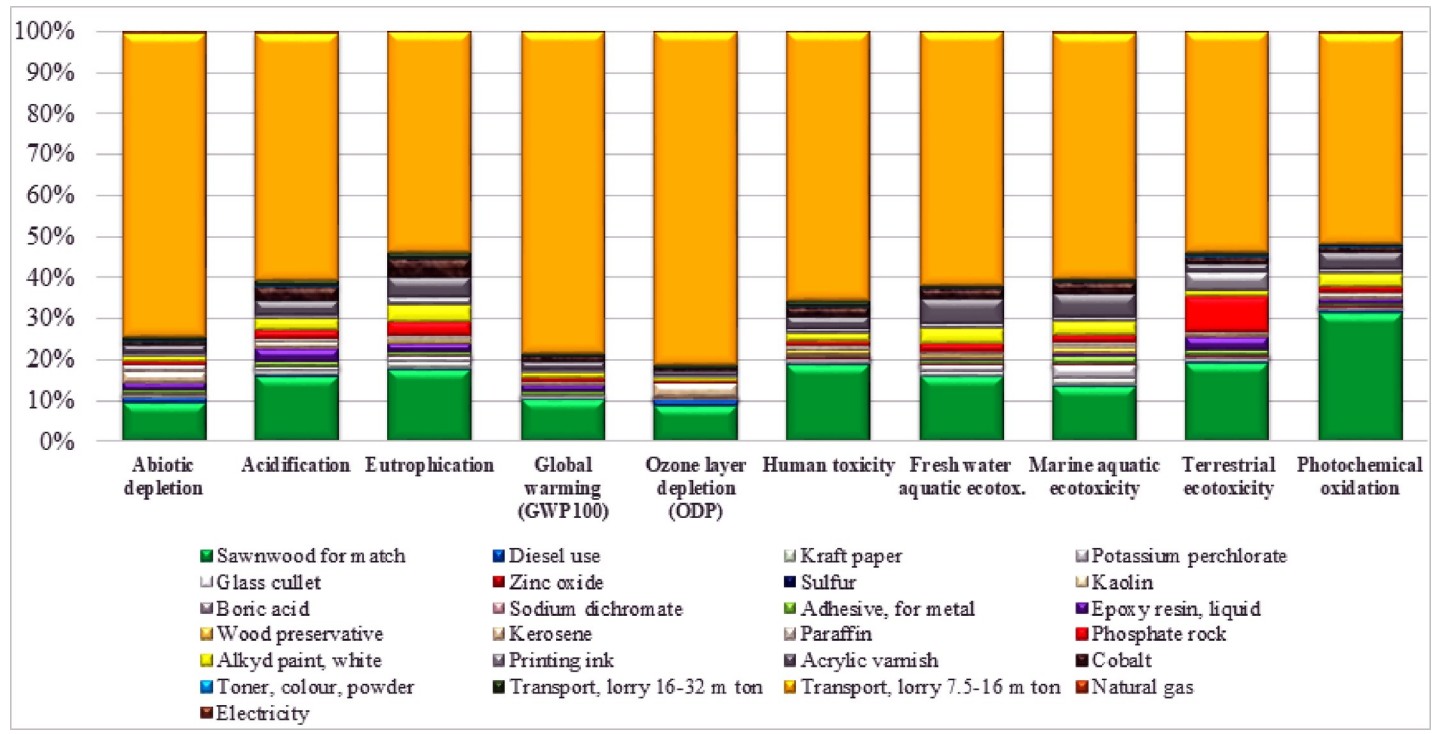

**Fig 3. Relative contribution per process (in %) to various environmental impacts.**

transport of primary materials and sawn wood were the major contributors (64% and 14%, respectively) to the marine aquatic ecotoxicity (MAE) impacts category. The acrylic varnish, dry paint (alkyd paint white), cobalt, and potassium perchlorate contribute (6%, 4%, 3%, and 3%, respectively), while red phosphorous (phosphate rock), paper (kraft paper), additionally pearl and SWD glue (adhesive) equally contribute (2%) to MAE impacts. The match produce was also responsible for most of the effects in terrestrial ecotoxicity (TE), mostly from the transport of primary resource material (58%), sawn wood for a match (21%). The red phosphorous (phosphate rock) and printing ink contribute (9% and 5%, respectively), while dry and resin gum (epoxy resin), acrylic varnish, and cobalt each also had contribution (3%, 2%, and 2%, respectively) to the TE impacts category.

## Carbon footprint, carbon stock, and net carbon flux

The quantity of organic carbon stored often considered in the cradle-to-gate assessments mainly for the embodied carbon may be released back to the atmosphere i.e. through burning [14, 24, 25]. Wood products are often considered as carbon neutral since they sequester carbon dioxide during the trees' growth equivalent to the emissions released during the final burning or decay [26, 27]. However, neutrality in biogenic carbon does not necessarily means greenhouse gas neutrality; since carbon emissions can occur in the form of methane ($CH_4$) and nitrous oxide ($N_2O$), more potent GHGs than $CO_2$ can be resultant from unsustainable forestry [28–30]. Carbon footprint can be defined as the amount of total GHG emissions directly or indirectly caused by a product, or process, generally calculated in the mass of carbon dioxide equivalents ($CO_2e$) [13, 31]. The percent average carbon content in the wood was measured at 52.4% of the mass, according to CORRIM guidelines for life cycle inventories on wood products [22, 32, 33]. Hence, the carbon content in the wood of one carton of a match is

38.35 kg in KP, Pakistan. In the assessment of carbon flux, the carbon stored in a wood-based product is measured as negative carbon, which is removed from the total carbon footprint of the product. Thus, the carbon store (38.35 kg CO2e) in one carton of a match can be used to offset the carbon footprint (43.69 kg CO2e), to evaluate the net carbon flux, which is +5.35 kg $CO_{2e}$ per carton of the match produced during the year 2019–2020. The fundamental assumption of carbon-neutrality of wood biomass relies upon forest management practices and product end-use, which was excluded from the present study. Pakistani forest management practices appear to be unsustainable at the present time. Pakistan's forest cover remains stagnant at less than 5%, and the deforestation rate is 2.1%, which is more noteworthy among all the South Asian countries [34]. Therefore, it is essential to implement forest management strategies to ensure sustainable raw wood materials to fulfil need of the match industry in KP, Pakistan. Reforestation and afforestation plan such as the "Ten Billion Tree Plantation Program" and"Green Clean Pakistan Program" of the Federal Government of Pakistan may support to confirm a sustainable wood supply in the future [35].

## Cumulative energy demand

The overall cumulative energy demand for producing one carton match was 715.860 MJ from three impact categories: non-renewable fossil, non-renewable biomass, and renewable water (Fig 4). Among the three types, non-renewable fossil had the maximum contribution (99%), while renewable water had a negligible contribution as can be seen in (Fig 5). Among the manufacturing activities, primary materials transport, sawn wood for a match, red phosphorous (phosphate rock), acrylic varnish, cobalt, and printing ink were the most energy-intensive activities, as can be seen in (Table 3).

## Emissions to air, soil, and water from one carton of match production

Emissions of hazardous substances to air, water, and soil for one carton of the match produced by match industries in KP during 2019–20 are summarized in S1-S3 Tables in S1 Appendix. Most of the emissions are released from the primary materials transport, followed by sawn wood for a match, red phosphorous, printing ink, acrylic varnish, cobalt, and dry paint (alkyd paint white). These substances largely contributed to air emission includes 1-Propanol (526.755 µg) followed by Iron (845.797 mg), Carbon dioxide (571.461 mg), Lactic acid (730.244 ng), Acetic acid (224.355 mg), Heat, waste (455.272 kJ), Ethane (411.800 mg) and Antimony (361.053 mg). While the substances which mainly contributed to water were Hydrogen chloride (990.573 mg) followed by Formic acid (471.262 ng), COD, Chemical Oxygen Demand (180.081 g), BOD5, Biological Oxygen Demand (164.118 g) and Ammonium, ion (155.828 mg). In contrast, the substances which mostly contributed to soil were Sodium (962.382 mg) followed by Nickel (469.052 µg), Selenium (413.727µg), Nitrogen (395.788 µg), Cobalt (289.361 µg), and Arsenic (236.611 µg). Nevertheless, none of the surveyed match manufacturing industries has emissions control devices. Therefore, match industries can cut their emissions to half by fixing emissions control devices such as fabric absorptive filters and oxidation systems.

## Specific hazardous substances contributed to various environmental impacts

Specific hazardous substances result in harmful effects causing various environmental impacts for one carton of the match produced by match industries in KP during 2019–20 are summarized in S4-S13 Tables in S1 Appendix. Most of the emissions were from the primary materials transport, followed by sawn wood for a match, red phosphorous (phosphate rock), printing

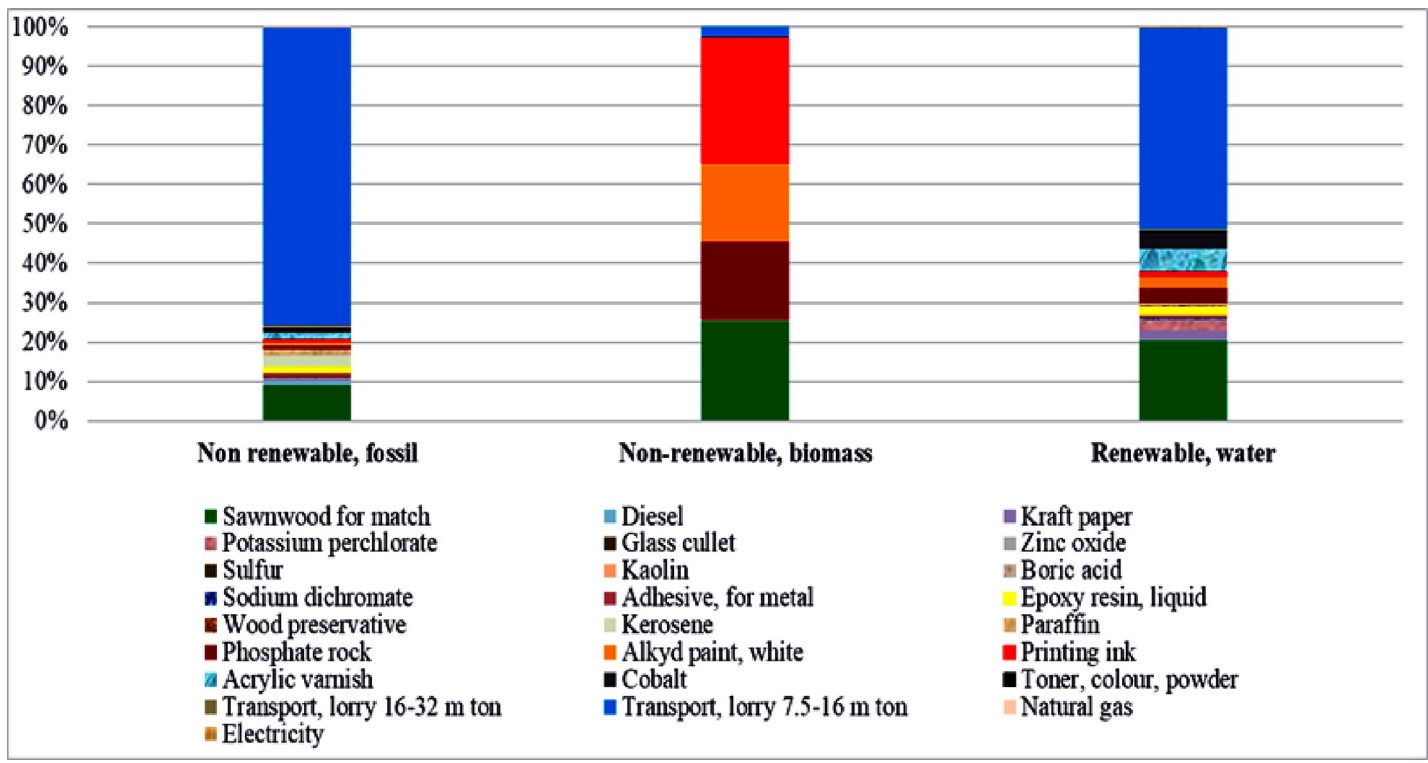

**Fig 4. Energy consumption calculated through Cumulative Energy Demand (CED) indicator.**

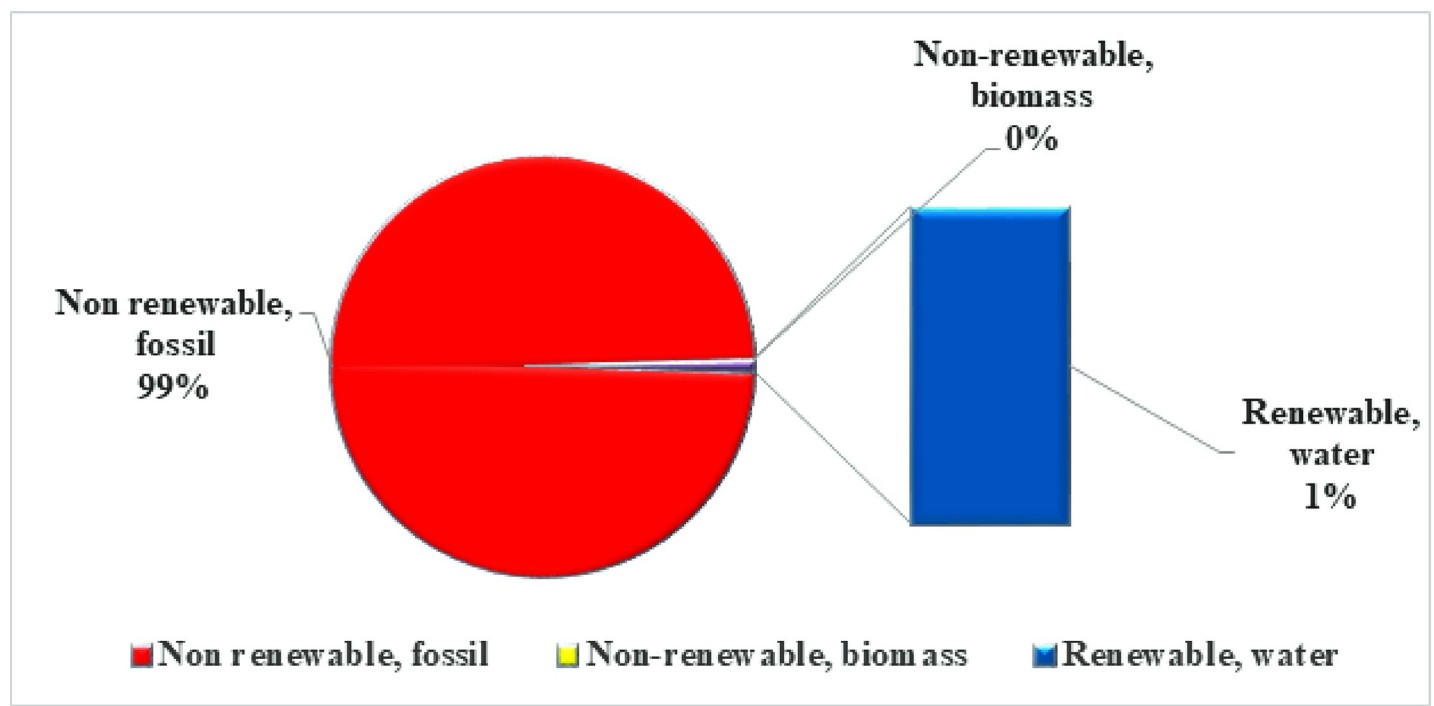

**Fig 5. Relative percent contribution of each category to total cumulative energy demand.**

Table 3. Summary of categories related to CED indicator and associated hotspots in match production.

| Impact category | Unit | Total | Most effective sectors/hotspots |
|---|---|---|---|
| Non-renewable, fossil | MJ | 708.979 | Primary materials transport, sawn wood for a match, red phosphorous (phosphate rock), printing ink |
| Non-renewable, biomass | MJ | 0.511 | Sawn wood for a match, printing ink, dry paint (alkyd paint white), red phosphorous (phosphate rock) |
| Renewable, water | MJ | 6.370 | Primary materials transport, sawn wood for a match, acrylic varnish, cobalt |
| Total | MJ | 715.860 | |

ink, acrylic varnish, cobalt, and dry paint (alkyd paint white). The substances which largely contributed to global warming potential were carbon dioxide, fossil (41.46060491 kg CO2 eq), Chloroform (3.28882E-06 kg CO2 eq), Methane (5.563E-07 kg CO2 eq), and Methane, dichloro-, HCC-30 (9.42686E-07 kg CO2 eq). The substances, which mostly contributed to abiotic depletion, were Cadmium (9.82531E-05 kg Sb eq), Bromine (5.81031E-08 kg Sb eq), Iodine (7.93993E-08 kg Sb eq), Iron (7.12297E-08 kg Sb eq), and Zinc (8.8627E-06 kg Sb eq). The substances, which mainly contributed to acidification, were Sulfur dioxide (0.0969 kg SO2 eq) and Nitrogen oxides (0.0432 kg SO2 eq). The substances which largely contributed to eutrophication were Phosphorus (7.44E-05 kg PO4— eq), Phosphorus (6.4292E-06 kg PO4— eq), and Ammonium carbonate (6.80352E-10 kg PO4— eq). The substances which mainly contributed to freshwater aquatic Ecotoxicity were Tin (6.55934E-05 kg 1,4-DB eq), Chloroform (4.56088E-11 kg 1,4-DB eq), and Antimony (4.13169E-07 kg 1,4-DB eq). The substances which mostly contributed to human health were Ethene (7.96334E-05 kg 1,4-DB eq), Mercury (7.56132E-06 kg 1,4-DB eq), and Cobalt (5.62642E-05 kg 1,4-DB eq). The substances which largely contributed to aquatic toxicity were Cobalt (633.1352516 kg 1,4-DB eq), Copper (419.92 kg 1,4-DB eq), and Nickel (1366.93 kg 1,4-DB eq). The substances which mainly contributed to Ozone layer depletion (OLD) were Methane, bromotrifluoro-, Halon 1301 (7.16E-06 kg CFC-11 eq) and Methane, trichlorofluoro CFC-11 (7.28559E-11 kg CFC-11 eq). The substances which largely contributed to photochemical oxidation were 1-Butanol (9.82014E-10 kg C2H4 eq), Heptane (8.14845E-05 kg C2H4 eq), and Methyl formate (7.84943E-13kg C2H4 eq), while the substances which mostly contributed to terrestrial toxicity were Ethene (9.68952E-17 kg 1,4-DB eq), Zinc (8.36815E-24 kg 1,4-DB eq) and Copper (7.31686E-24 kg 1,4-DB eq).

## Water footprint or water scarcity index (WSI)

The results of water footprint or water scarcity index for one carton of match production and its relative contribution per process by match industries in KP during 2019–20 are presented in (Fig 6) and (Table 4). The total water scarcity index or water footprint for manufacturing one carton match was calculated as 0.265332 m³. Among the different manufacturing activities, the transport of primary materials, sawn wood for matchsticks, and acrylic varnish had the highest contributions to the water scarcity index whereas dry, and resin gum (epoxy resin), cobalt, red phosphorous (phosphate rock), dry paint (alkyd paint white) and kerosene oil had minor contribution to water scarcity index.

## Sensitivity analysis for clean and green option in matchstick manufacture process

The sensitivity analysis aims to identify those inputs and outputs, which have significant environmental impacts and then how those can be reduced to minimize their perilous

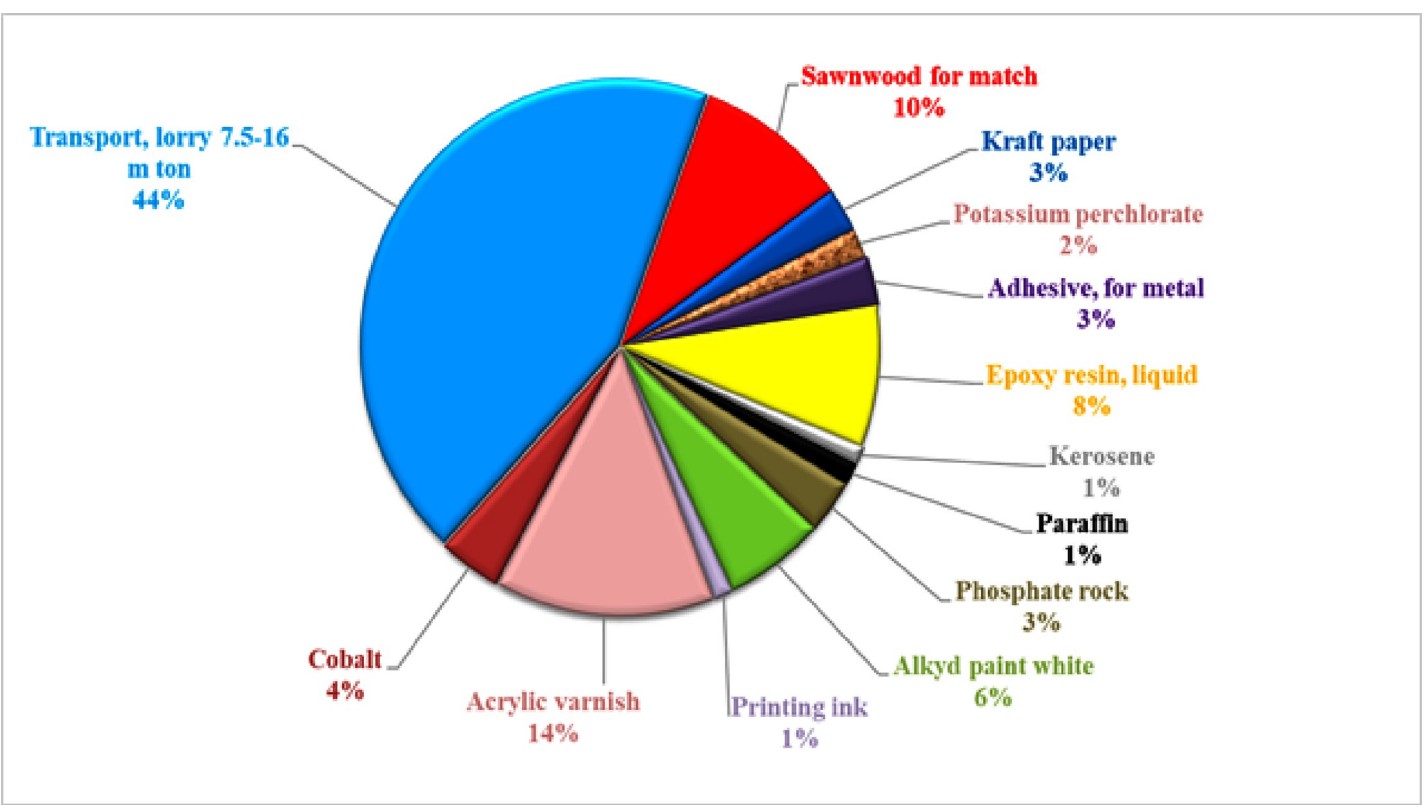

**Fig 6. Relative contribution per process (in %) to water scarcity index**

environmental impacts [36, 37]. A cradle-to-gate LCA of match manufacture was conducted in KP, Pakistan. The main hotspots sources of match manufacture process in KP, Pakistan are summarized in (Table 5). From the results, it is evident to conclude that the hotspot source in the match production process was primary material (wood logs) transport. Therefore, with a 20% reduction in the direct material transport from the baseline value could reduce the environmental burdens such as abiotic depletion (15%), acidification (12%), eutrophication (11%), GWP (16%), OLD (16%), human toxicity (13%), freshwater aquatic Ecotoxicity (12%), marine aquatic Ecotoxicity (12%), terrestrial Ecotoxicity (0.84%) and photochemical oxidation (10%). Moreover, 30% reductions of primary material transport also reduced the environmental burdens, such as abiotic depletion (22%), acidification (18%), eutrophication (16%), GWP (24%), OLD (24%), human toxicity (20%), freshwater aquatic Ecotoxicity (19%), marine aquatic Ecotoxicity (18%), terrestrial Ecotoxicity (16%) and photochemical oxidation (15.50%). The same as with 20% reduction in cumulative energy demand from the baseline value, so that the decrease in the value of cumulative energy demand reduced the energy consumption, such as nonrenewable fossil (15%), nonrenewable biomass (0.39%), and renewable water (10%). In particular, a 30% reduction of cumulative energy demand also reduced energy consumption, such as nonrenewable fossil (23%), nonrenewable biomass (0.78%), and renewable water (15%), as shown in (Table 6).

## Discussion

The hotspot sources responsible for health issues due to one carton of match production by match industries in KP during 2019–2020 were sawn wood, red phosphorous (phosphate

**Table 4. Water scarcity index (WSI) of one carton match production in KP province.**

| Impact category | WSI |
| --- | --- |
| Sawn wood | 0.0248 |
| Diesel | 0.0011 |
| Kraft paper | 0.0070 |
| Potassium perchlorate | 0.0047 |
| Glass cullet | 1.6137 |
| Zinc oxide | 0.0001 |
| Sulfur | 9.7853 |
| Kaolin | 0.0001 |
| Boric acid | 0.0001 |
| Sodium dichromate | 4.6927 |
| Adhesive, for metal | 0.0077 |
| Epoxy resin | 0.0221 |
| Wood preservative | 0.0007 |
| Kerosene | 0.0027 |
| Paraffin | 0.0034 |
| Phosphate rock | 0.0084 |
| Alkyd paint | 0.0163 |
| Printing ink | 0.0031 |
| Acrylic varnish | 0.0356 |
| Cobalt | 0.0101 |
| Toner, colour, powder | 0.0005 |
| Electricity | 8.4393 |
| Transport, lorry 7.5–16 m ton | 0.1143 |
| Natural gas | 6.9434 |

rock), cobalt, transport of primary materials, dry and resin gum (epoxy resin), acrylic varnish, dry paint (alkyd paint white) and kerosene oil. In particular, the sawn wood produces fine dust particles, which may influence mucus production and induce adverse effects on the skin [38]. Wood dust can cause both Type I allergy (Immediate Hypersensitivity (Anaphylactic Reaction)) and Type IV allergy (Cell-Mediated (Delayed Hypersensitivity)). In most cases, allergy develops because of Type IV allergy. Red Phosphorus has more damaging oxidizing effects. Moreover, in some instances the simple act of the interaction itself or the reaction's product may determine an abrupt fire or explosion [39]. Burning red phosphorus produces tetra phosphorus dioxide as smoke, which could be destructive for the eyes, skin, and respiratory tract [40]. Cobalt used in the match manufacturing industry is also toxic, and it has been associated with the outbreak of challenging metal disease, asthma, and allergies in the match industry workers [41]. Diesel engines of primary transportation have widespread usage in KP Pakistan in comparison with gasoline engines due to their low operating costs, energy efficiency, high stability and consistency [42]. Diesel engine based vehicles are the prime source of freight in KP, Pakistan. Even though diesel engine vehicles have many points of interest, but these vehicles causes significant environmental pollution. Diesel exhaust gas contains an advanced measure of particulate matter (PM) and $NO_X$ emissions liable for severe environmental health [43]. Diesel utilized for transport purposes influences human wellbeing antagonistically, causes acid rains, ground-level ozone, and diminishes visibility. Studies have indicated that the inhalation of diesel exhaust gas causes lung harm and respiratory issues, and this is evident from previous research studies that diesel vehicles emissions may cause cancer in humans [44]. The

**Table 5.  Comparative environmental impacts assessment of baseline results with the results obtained by 20% and 30% reductions in the primary material (wood logs) transport, respectively.**

| | Impact category | Abiotic depletion | Acidification | Eutrophication | Global warming potential | Ozone layer depletion | Human toxicity | Fresh water aquatic ecotoxicity | Marine aquatic ecotoxicity | Terrestrial ecotoxicity | Photochemical oxidation |
|---|---|---|---|---|---|---|---|---|---|---|---|
| | Unit | kg Sb eq | kg SO2 eq | kg PO4- eq | kg CO2 eq | kg CFC-11 eq | kg 1,4-DB eq | kg 1,4-DB eq | kg 1,4-DB eq | kg 1,4-DB eq | kg C2H4 eq |
| Primary material (wood logs) Transport | Baseline value of primary transport use (160.4 t.km) | 0.33 | 0.142 | 0.033 | 43.69 | 7.55E-0 | 17.844 | 7.1943 | 11758.8 | 0.099 | 0.0104 |
| | 20% reduction in primary transport use (128 t.km) | 0.2811 | 0.124 | 0.030 | 36.78 | 6.32E-0 | 15.481 | 6.3000 | 10335.9 | 0.088 | 0.0094 |
| | Percent decrease in overall environmental impacts | 15.00% | 12.00% | 11.00% | 16.00% | 16.00% | 13.00% | 12.00% | 12.00% | 0.84% | 10.00% |
| Primary material (wood logs) Transport | Baseline value of primary transport use (160.4 t.km) | 0.3307 | 0.142 | 0.0338 | 43.69 | 7.55E-0 | 17.844 | 7.1943 | 11758.8 | 0.099 | 0.0104 |
| | 30% reduction in primary transport use (112 t.km) | 0.2567 | 0.116 | 0.0283 | 33.36 | 5.71E-0 | 14.314 | 5.8584 | 9633.36 | 0.083 | 0.0088 |
| | Percent decrease in overall environmental impacts | 22.00% | 18.00% | 16.00% | 24.00% | 24.00% | 20.00% | 19.00% | 18.00% | 16.00% | 15.50% |

kerosene oil-primarily causes pulmonary problems, including chemical pneumonitis, through the central nervous system. Moreover, children can be intoxicated by consumption, inhalation, or skin contact with kerosene oil [45].

## Limitations of the present study

The current study is based on cradle-to-gate life cycle analysis, which excluded a portion of possibly significant causes of emissions from match-producing industries because of the unavailability of accurate and precise data. For example, forest operations such as planting,

**Table 6.  Comparative energy consumptions of baseline results with the results obtained by 20% and 30% reductions in cumulative energy demand respectively.**

| | Impact category | Non-renewable, fossil | Non-renewable, biomass | Renewable, water |
|---|---|---|---|---|
| | Unit | MJ | MJ | MJ |
| Cumulative energy demand | The baseline value of cumulative energy demand (160.4 t.km) | 708.979 | 0.511 | 6.370 |
| | 20% reduction in cumulative energy demand (128t.km) | 601.052 | 0.509 | 5.715 |
| | Percent decrease in overall energy consumption | 15.00% | 0.39% | 10.00% |
| Cumulative energy demand | The baseline value of cumulative energy demand (160.4 t.km) | 708.979 | 0.511 | 6.370 |
| | 30% reduction in cumulative energy demand (112t.km) | 547.755 | 0.507 | 5.391 |
| | Percent decrease in overall energy consumptions | 23.00% | 0.78% | 15.00% |

seedlings, fertilizer use, thinning, site preparation, and final harvesting [46]. Nevertheless, several research studies [47–54] have reported the impacts of forest operations are exceptionally derisory in correlation with the production process. The matchsticks' utilization and removal were likewise excluded from this study, based upon the fact that a large portion is consumed inside the country and exported to Afghanistan from Pakistan. Essentially, LCA gives an all-encompassing perspective on environmental impacts, but it did not address the systems' potent effects. It does not think about varieties in a global context too. Hence, environmental impacts are not space-specific and time-specific or results from LCA give potential environmental impacts. The study's precision and consistency significantly rely upon the worth and availability of precise real data. The geographic extent of databases used for environmental impacts modeling in the present study is restricted to Europe and USA, effecting the comparison between studies conducted in other parts of the world such as developing countries like Pakistan. Hence, country-specific databases should be developed just like Ecoinvent in Europe and USLCI and CORRIM in the USA, which possess relevant, precise, reliable, and comprehensive data for these countries' local perspective. Besides, the current LCA study emphasis only on the environmental burdens of match manufacture and does not integrate social or economic impacts of the match product; nevertheless, it should be examined in future studies through Life Cycle Costing (LCC) and Social Life Cycle Assessment (SLCA) of the match industry in Pakistan.

## Conclusions and recommendations

This study presented the environmental impacts, water footprint and cumulative energy demand from resource inputs and outputs such as primary materials transport, sawn wood for a match, red phosphorous (phosphate rock), printing ink, dry paint (alkyd paint white), acrylic varnish, cobalt, and kerosene oil for one carton match produced in KP, Pakistan. The results exhibited that the total water scarcity index or water footprint for manufacturing one carton match was 0.265332 $m^{3,}$ whereas the total cumulative energy demand for manufacturing one carton match was (715.860 MJ), wholly obtained from nonrenewable fossil sources (708.979 MJ). The transport of primary materials, sawn wood for a matchstick, cobalt, red phosphorous (phosphate rock), dry and resin gum (epoxy resin), acrylic varnish, dry paint (alkyd paint white), and kerosene oil had maximum contributions to environmental impact categories. In contrast, transport of primary materials and sawn wood for a match contributed significantly to global warming (GWP 100), abiotic depletion, photochemical oxidation, ozone depletion, eutrophication potential, human toxicity, and aquatic ecotoxicity impacts. A sensitivity analysis was conducted to identify primary material (wood logs) transport as well as for cumulative energy demand. With a 20% reduction of primary material transport for the baseline value, the decrease in the value of direct material transport reduced the environmental burdens such as abiotic depletion (15%), acidification (12%), eutrophication (11%), GWP (16%), OLD (16%), human toxicity (13%), freshwater aquatic Ecotoxicity (12%), marine aquatic Ecotoxicity (12%), terrestrial Ecotoxicity (0.84%) and photochemical oxidation (10%). While a 30% reductions of primary material transport also reduced the environmental burdens such as abiotic depletion (22%), acidification (18%), eutrophication (16%), GWP (24%), OLD (24%), human toxicity (20%), freshwater aquatic Ecotoxicity (19%), marine aquatic Ecotoxicity (18%), terrestrial Ecotoxicity (16%) and photochemical oxidation (15.50%). With 20% reductions in cumulative energy demand concerning the baseline value, the decrease in the value of incremental energy demand reduced energy consumptions, such as nonrenewable, fossil (15%), nonrenewable, biomass (0.39%), and renewable, water (10%). However, with a 30% reduction of cumulative energy demand, energy consumption was reduced as well, with the following values:

nonrenewable, fossil (23%), nonrenewable, biomass (0.78%), and renewable, water (15%). This study is the ever-first study on LCA of matchsticks manufacturing industries in Pakistan and all over the world, and hence, it can be a baseline for other wood based and petro-based industries. The primary material (wood logs) transportation can be minimized by diverting match industries freight from indigenous routes to high mobility highways. Heavy vehicles in Pakistan often prefer long and local roads to avoid tax on toll and avoid load restrictions. In addition, promote the industrial forestry near the industrial zones to overcome the primary material (wood logs) transportation from far away regions to the match manufacturing units causing environmental impacts. None of the surveyed match manufacturing industries have installed pollution control systems; however, if pollution control systems were installed, there could be a significant decrease in pollution caused by match production in KP, Pakistan.

## Supporting information

**S1 Appendix.**
(DOCX)

## Acknowledgments

The authors are thankful for the matchsticks industries and their managerial staff for their contribution to data collection at the Hayatabad Industrial Estate, District Peshawar, Hattar Industrial Estate, District Haripur, and Special Industrial Zone Risalpur, District Nowshera of Khyber Pakhtunkhwa province of Pakistan. Any findings, conclusions, and recommendations expressed in this article belong to the authors and do not reflect the contributing bodies' views.

## Author Contributions

**Conceptualization:** Syeda Asma Bano.

**Data curation:** Syeda Asma Bano.

**Formal analysis:** Sher Shah.

**Investigation:** Ume Habiba.

**Methodology:** Maimoona Sabir.

**Project administration:** Andleeb Akhtar.

**Resources:** Andleeb Akhtar.

**Software:** Sher Shah.

**Supervision:** Majid Hussain.

**Validation:** Ayesha Shoukat.

**Visualization:** Ayesha Shoukat.

**Writing – original draft:** Najeeb Ullah, Majid Hussain.

**Writing – review & editing:** Ume Habiba, Samreen Ramzan, Muhammad Israr, Syed Moazzam Nizami, Majid Hussain.

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
