## [Decision Letter · Decision Letter 0]

8 Apr 2021

PONE-D-21-08698

Environmental impacts, water footprint and cumulative energy demand of match industry in Pakistan

PLOS ONE

Dear Dr. Hussain,

Thank you for submitting your manuscript to PLOS ONE. After careful consideration, we feel that it has merit but does not fully meet PLOS ONE’s publication criteria as it currently stands. Therefore, we invite you to submit a revised version of the manuscript that addresses the points raised during the review process.

We look forward to receiving your revised manuscript.

Kind regards,

Bing Xue, Ph.D.

Academic Editor

PLOS ONE

Journal Requirements:

1. Please ensure that your manuscript meets PLOS ONE's style requirements, including those for file naming. The PLOS ONE style templates can be found athttps://journals.plos.org/plosone/s/file?id=wjVg/PLOSOne_formatting_sample_main_body.pdf and https://journals.plos.org/plosone/s/file?id=ba62/PLOSOne_formatting_sample_title_authors_affiliations.pdf

Additional Editor Comments (if provided):

The paper is interesting, but some modification should be conducted before it was considered for publication:

1) the introduction should be improved by adding more new literature, and please also try to outline your research question and potential contributions.

2) please improve the discussion section

3) the language should be improved.

Reviewers' comments:

Reviewer's Responses to Questions

**Comments to the Author**

1. Is the manuscript technically sound, and do the data support the conclusions?

Reviewer #1: Yes

2. Has the statistical analysis been performed appropriately and rigorously? 

Reviewer #1: Yes

3. Have the authors made all data underlying the findings in their manuscript fully available?

Reviewer #1: No

4. Is the manuscript presented in an intelligible fashion and written in standard English?

Reviewer #1: Yes

5. Review Comments to the Author

Reviewer #1: The article is accepted but before the publication of this article the author needs to make the required changes as explained above in the comments. The study contains an important factor or outcome which is "the innovation" which is not only important but an essential element for any organization seeking to sustain. Methodology of the paper is fit to data, and analysis done by the authors is according to the requirements of paper title.

6. PLOS authors have the option to publish the peer review history of their article (what does this mean?). If published, this will include your full peer review and any attached files.

Reviewer #1: No

---

## [Author Response · Author response to Decision Letter 0]

28 Apr 2021

Response to Reviewers

PONE-D-21-08698

Environmental impacts, water footprint and cumulative energy demand of match industry in 

We look forward to receiving your revised manuscript.

Kind regards,

Bing Xue, Ph.D.

Academic Editor

PLOS ONE

Journal Requirements:

Answer: Revised manuscript is formatted according to PloS One style now. 

2. We suggest you thoroughly copyedit your manuscript for language usage, spelling, and grammar. 

Answer: The revised manuscript is thoroughly copyedited for language usage, spelling and grammar by one of my English native friend Dr. Alessandra Serafino. 

3. Please include captions for your Supporting Information files at the end of your manuscript, and update any in-text citations to match accordingly

Answer: Captions are provided for supporting information files at the end of the revised manuscript and all the citation and references are update and matched. 

Answer: all the citations and references are double checked and corrected in the revised manuscript as per PLOS One author guidelines. 

1) the introduction should be improved by adding more new literature, and please also try to outline your research question and potential contributions.

Answer: the introduction part of the manuscript is revised and improved. 

2) please improve the discussion section

Answer: Discussion section is improved in the revised manuscript. 

3) the language should be improved.

Answer: Native English Friend improves the English language of the manuscript voluntarily. 

1. Is the manuscript technically sound, and do the data support the conclusions?

Reviewer #1: Yes

2. Has the statistical analysis been performed appropriately and rigorously? 

Reviewer #1: Yes

3. Have the authors made all data underlying the findings in their manuscript fully available?

Reviewer #1: No

All the data underlying the findings are attached as supplemental materials in the revised manuscript. 

4. Is the manuscript presented in an intelligible fashion and written in standard English?

Reviewer #1: Yes

4. Review Comments to the Author

Reviewer #1: The article is accepted but before the publication of this article the author needs to make the required changes as explained above in the comments. The study contains an important factor or outcome, which is "the innovation" which is not only important but an essential element for any organization seeking to sustain. Methodology of the paper is fit to data, and analysis done by the authors is according to the requirements of paper title.

Answer: Thank you very much respected Reviewer # 1 for your positive remarks and recommendation of acceptance of our paper. 

---

## [Decision Letter · Decision Letter 1]

6 May 2021

Environmental impacts, water footprint and cumulative energy demand of match industry in Pakistan

PONE-D-21-08698R1

Dear Dr. Hussain,

We’re pleased to inform you that your manuscript has been judged scientifically suitable for publication and will be formally accepted for publication once it meets all outstanding technical requirements.

Kind regards,

Bing Xue, Ph.D.

Academic Editor

PLOS ONE

Additional Editor Comments (optional):

Reviewers' comments:

Reviewer's Responses to Questions

**Comments to the Author**

1. If the authors have adequately addressed your comments raised in a previous round of review and you feel that this manuscript is now acceptable for publication, you may indicate that here to bypass the “Comments to the Author” section, enter your conflict of interest statement in the “Confidential to Editor” section, and submit your "Accept" recommendation.

Reviewer #1: All comments have been addressed

2. Is the manuscript technically sound, and do the data support the conclusions?

Reviewer #1: Yes

3. Has the statistical analysis been performed appropriately and rigorously? 

Reviewer #1: Yes

4. Have the authors made all data underlying the findings in their manuscript fully available?

Reviewer #1: Yes

5. Is the manuscript presented in an intelligible fashion and written in standard English?

Reviewer #1: Yes

6. Review Comments to the Author

Reviewer #1: the author did a good work , he fullfilled all the requirements according to journal standard. The methodology part is uniqe, the data analysis techniques are well defined and results calculation is according to theoretical description of paper.

7. PLOS authors have the option to publish the peer review history of their article (what does this mean?). If published, this will include your full peer review and any attached files.

Reviewer #1: **Yes: **Muneeb Ahmad

---

## [Editor Report · Acceptance letter]

12 May 2021

PONE-D-21-08698R1 

Environmental impacts, water footprint and cumulative energy demand of match industry in Pakistan 

Dear Dr. Hussain:

I'm pleased to inform you that your manuscript has been deemed suitable for publication in PLOS ONE. Congratulations! Your manuscript is now with our production department. 

Kind regards, 

on behalf of

Professor Bing Xue 

Academic Editor

PLOS ONE